# Effect of In-Mold Annealing on the Properties of Asymmetric Poly(l-lactide)/Poly(d-lactide) Blends Incorporated with Nanohydroxyapatite

**DOI:** 10.3390/polym13162835

**Published:** 2021-08-23

**Authors:** Martin Boruvka, Cenek Cermak, Lubos Behalek, Pavel Brdlik

**Affiliations:** Department of Engineering Technology, Faculty of Mechanical Engineering, Technical University of Liberec, Studenstka 2, 461 17 Liberec, Czech Republic; cenek.cermak@tul.cz (C.C.); lubos.behalek@tul.cz (L.B.); pavel.brdlik@tul.cz (P.B.)

**Keywords:** poly(l-lactide), poly(d-lactide), stereocomplex, nanohydroxyapatite, nanocomposites, mechanical properties, impact resistance, heat resistance, crystallization

## Abstract

The proper choice of a material system for bioresorbable synthetic bone graft substitutes imposes strict requirements for mechanical properties, bioactivity, biocompatibility, and osteoconductivity. This study aims to characterize the effect of in-mold annealing on the properties of nanocomposite systems based on asymmetric poly(l-lactide) (PLLA)/Poly(d-lactide) (PDLA) blends at 5 wt.% PDLA loading, which was incorporated with nano-hydroxyapatite (HA) at various concentrations (1, 5, 10, 15 wt.%). Samples were melt-blended and injection molded into “cold” mold (50 °C) and hot mold (100 °C). The results showed that the tensile modulus, crystallinity, and thermal-resistance were enhanced with increasing content of HA and blending with 5 wt.% of PDLA. In-mold annealing further improved the properties mentioned above by achieving a higher degree of crystallinity. In-mold annealed PLLA/5PDLA/15HA samples showed an increase of crystallinity by ~59%, tensile modulus by ~28%, and VST by ~44% when compared to neat hot molded PLLA. On the other hand, the % elongation values at break as well as tensile strength of the PLLA and asymmetric nanocomposites were lowered with increasing HA content and in-mold annealing. Moreover, in-mold annealing of asymmetric blends and related nanocomposites caused the embrittlement of material systems. Impact toughness, when compared to neat cold molded PLLA, was improved by ~44% with in-mold annealing of PLLA/1HA. Furthermore, fracture morphology revealed fine dispersion and distribution of HA at 1 wt.% concentration. On the other hand, HA at a high concentration of 15 wt.% show agglomerates that worked as stress concentrators during impact loading.

## 1. Introduction

As global society continues to grow, sustainability issues concerning our raw material systems arise as well. Sustainability coupled with the rising environmental concerns and fossil resources depletion present considerable challenges for the polymer industry [1]. To alleviate these problems, governments in many countries are enacting laws that encourage the use of recycled, renewable, and biodegradable polymers [2]. Particularly in Europe, topics such as greenhouse gas emission, CO_2_ neutrality, renewability, embodied energy, toxicity, and resource depletion are driven by regulations. 

At the same time, new sustainability platforms arise as well. An example is Think 2030, which is based at the Institute for European Environmental Policy (IEEP) and will produce science-based agenda recommendations for European environmental policy beyond 2020 [3].

Biobased polymers have been attracting attention as renewable materials that can replace conventional polymers, which are synthesized from unsustainable fossil based resources [4]. Among these, polylactide (PLA) has exhibited vast appeal in the past decades due to its good mechanical properties, renewability, biodegradability, and biocompatibility. The price of PLA is only slightly higher compared to the average commercial polymers and is likely to decline further with increasing demand [5]. PLA belongs to the family of compostable aliphatic polyesters which are derived from renewable biomass resources. Lactic acid (LA), which is the simplest α-hydroxy acid, is the basic building block of PLA. Due to a chiral carbon atom of LA, PLA has two stereoisomeric forms: poly(l-lactide) (PLLA) and poly(d-lactide) (PDLA) [6]. Materials with variable optical and physio-chemical properties can be produced via controlling the proportion of the l- and d-enantiomers. This allows the production of relatively wide spectra of stereoisomeric PLA forms to match performance requirements [7]. Despite this, the application possibilities of PLA have been still limited notably due to slow crystallization rate, low heat, and impact resistance. Substantial efforts to address these issues have been undertaken to improve property deficiency. To achieve this, research is being carried out to develop modified PLA systems by blending, plasticization, copolymerization, and the addition of different types of fillers [8].

One of the most effective methods to enhance heat resistance, thermal stability, mechanical performance, and hydrolysis resistance of PLA is the formation of stereocomplex (SC) crystallites between enantiomeric PLLA and PDLA [9,10,11]. This unique idea was at first reported by Ikeda et al. [12] in 1987. Since then, numerous studies have been made on PLA stereocomplex due to its higher thermal and mechanical properties [13,14,15,16]. Furthermore, improved rheology and crystallization kinetics have been reported by Yamane et al. [17], Inkinen et al. [18], and Shi et al. [19] through the investigation of asymmetric PLLA/PDLA blends. Wei et al. [20] revealed at his research by self-nucleation experiments and polarized light microscopy (POM) observations, that the highest nucleation density of PLLA/PDLA blends was achieved through incorporation of PDLA with a concentration of 5 wt.%. Subsequently, Wang et al. [21] observed the formation of a highly interconnected honeycomb network of SC crystallites through selective dissolution of asymmetric PLLA/PDLA blends with high-molecular-weight PDLA as a minor component. Pronounced crystallization of PLLA due to enhanced nucleation was observed at 5 wt.% loading of PDLA. 

Since the 1950s, PLA has been widely investigated for applications such as drug delivery systems, surgical sutures, three-dimensional scaffolds for tissue engineering, bone implants, and bone fixation devices [22]. Despite its good biocompatibility and bioresorbability, insufficient mechanical strength of pure PLA limits its use in applications such as bone tissue engineering [23]. Furthermore, PLA is not osteoconductive and lacks bone-bonding ability such as bioactive ceramics and glasses [24]. To address this issue, nano-inorganic fillers [25,26,27,28] were introduced into biopolymers to prepare composites which could mimic the structure and function of the extracellular matrix (ECM) and support cell adhesion, proliferation, differentiation [29]. Among these nanofillers, hydroxyapatite (HA) (Ca_10_(PO_4_)_6_(OH)_2_) has been used due to its excellent bioactivity, biocompatibility, and osteoconductivity [30,31,32]. Despite their several benefits, PLA/HA biocomposites often present some shortcomings in terms of performance [33]. HA nanoparticles tend to aggregate in PLA biocomposites because the surface energy of HA is much higher than that of PLA. Šupová [34] discussed several approaches to overcome the dispersion problem of hydroxyapatite particles in polymer matrices. Therefore, recent research have focused on improving the mechanical and thermal properties of PLA/HA biocomposites through surface and impact modifications [35,36]. Furthermore, the interfacial adhesion of PLA/HA has been improved by grafting PLA to HA through ring-opening polymerization (ROP) of lactide [37,38,39]. Surface modification of HA has been conducted by Akindoyo et al. [33] through incorporation of 2 wt.% dicopper hydroxide phosphate (Fabulase 361). Shuai et al. [40] used 2-carboxyethylphosphonic acid (CEPA) coupling agent to modify the surface of HA particles. Homogeneous dispersion of CEPA modified HA(C-HA) in PLA matrix with improved interfacial interaction was observed in the study. On the other hand, Ferri et al. [41] reported that well performed melt extrusion and subsequent injection molding of PLA/HA biocomposites (10–30 wt.% HA) enhanced thermo-mechanical properties of resulted nanocomposites. Nevertheless, increasing the HA content also resulted in reduced ability for energy absorption at impact conditions.

The primary objective of this study was to examine the influence of nanohydroxyapatite (HA) at various concentrations on the properties of asymmetric PLLA/PDLA blends. The novelty of this work is the study of the in-mold annealing effect of on the crystallization, mechanical properties, fractured morphology, and temperature and impact resistance of injection molded nanocomposites. The work also explores the usefulness of prolonged melt-blending and with subsequent injection molding techniques to manufacture plastic parts for bioresorbable synthetic bone graft substitutes. Furthermore, the influence of specific stereocomplex interactions on the dispersion and distribution of HA within asymmetric blends is discussed. 

## 2. Materials and Methods

### 2.1. Materials

The PLA grades used in this study were purchased in the form of granules from Total Corbion PLA (Gorinchem, Netherlands). As a base matrix we used Luminy L130 (≥99% L-isomer), and Luminy D070 (≥99% D-isomer) was used as a nucleating agent for stereocomplexation. Onwards from here both will be abbreviated as PLLA and PDLA, respectively. Hydroxyapatite powder under the trade name CA-PATOH-018-NP with an average particle size less than 100 nm and stereochemical purity of 98.5% was purchased from American Elements (Los Angeles, California, USA). Nanohydroxyapatite will be from here abbreviated as HA.

### 2.2. Samples Preparation

Neat PLLA, asymmetric PLLA/PDLA blends, PLLA/HA nanocomposites, and PLLA/PDLA/HA nanocomposites were prepared through melt-blending using MC 15 HT (Xplore, Sittard, Netherlands) microcompounder. Polymer granules and HA powder were dried in a VDB3 vacuum oven (Binder, Tuttlingen, Germany) at 80 °C for 12 h, prior to processing. Dispersion and distribution of HA has been controlled through stabilization of double screw torque. Samples were melt-blended for at least 5 min (depending on HA concentration) using built-in recirculation channel with a pair of conical screws (set to 100 rpm). Melting chamber temperatures has been set as constant 200 °C for PLLA based samples and 240 °C for PLLA/5PDLA based samples. After stabilizing the torque value (after at least 5 min), the recirculation valve has been switched and the homogenized melt was forced into the portable heated chamber of the injection molding machine IM12 (Xplore, Sittard, Netherlands). The chamber of the injection molding machine has been set to a temperature 5 °C higher than the microcompounder melting chamber. The injection process was divided into two parts; the first was an injection into a cold mold (50 °C) and the second was injection into a hot mold (100 °C). We are aware that term “cold mold” is not appropriate for temperatures as high as 50 °C, however it is used just for simplification reasons. Injection molds without cooling systems are regularly heated to a specific temperature after several cycles. Due to this we chose 50 °C as a base temperature. So, the processes differ only in the temperature of the mold and in the time after which the mold was removed from the machine, and the test specimens were released. The samples from cold mold were released immediately after injection molding, and samples from hot mold (100 °C) were allowed to crystallize within the mold for a further 120 s before demolding. During the injection molds were used to prepare standardized dumbbell tensile samples of 1B type according to the ISO 178 standard and test samples with dimensions of 80 × 10 × 4 mm^3^ according to the ISO 180 standard. The final concentrations of injection molded samples are listed in Table 1.

### 2.3. Differential Scanning Calorimetry (DSC)

Non-isothermal crystallization of the samples was characterized using DSC 1/700 (Mettler Toledo, Greifensee, SWITZERLAND). Measurements were performed on the middle part of tensile sample cross-sections (8 ± 0.5 mg), which were prepared by RM 2255 microtome (Leica, Nußloch, GERMANY). Samples were heated from 0 °C to 200 °C (PLLA/PDLA based samples up to 240 °C) at a 10 °C∙min^−1^ heating rate, then kept isothermal for 3 min to remove previous thermal history, and then cooled back to 0 °C under 10 °C∙min^−1^ cooling rate to observe melt crystallization. Analysis was performed under nitrogen flow rate of 50 ml∙min^−1^. The following were recorded from the first heating phase: glass transition temperature (*T*_g_); cold crystallization temperature (*T*_cc_) and enthalpy (Δ*H*_cc_); pre-melting recrystallization temperature (*T*_rc_) and enthalpy (Δ*H*_rc_); homocrystallite melting temperature (*T*_hm_) and enthalpy (Δ*H*_hm_); stereocomplex crystallite melting temperature (*T*_scm_) and enthalpy (Δ*H*_scm_). Parameters of the above-mentioned temperatures and enthalpies were taken as the peak temperatures and the areas of the melting endotherms or crystallization exotherms, respectively.

Crystallinity degrees (*χ*_c_) of neat PLLA and PLLA/HA nanocomposites were calculated as follows [42]:(1)$$\chi_{c\;}=\frac{\Delta H_{hm}-\Delta H_{cc}-\Delta H_{rc}}{\Delta H_{hm}^{0}\cdot W_{m}}\cdot 100\left[\%\right]$$
where Δ*H*^0^_hm_ is the melting enthalpy of 100% crystallized PLLA (106 J∙g^−1^) [43] and *W*_m_ is the weight fraction of PLLA.

Crystallinity degree (*χ*_c_) of asymmetric PLLA/PDLA blends and PLLA/PDLA/HA nanocomposites was calculated as follows [44]:(2)$$\chi_{c\;}=\frac{\Delta H_{hm}+\Delta H_{scm}-\Delta H_{cc}-\Delta H_{rc}}{\Delta H_{\left(h+sc\right)m}^{0}\cdot W_{m}}\cdot 100\;\;\;\left[\%\right]$$
where Δ*H*^0^_(h+sc)m_ is a calculated melting enthalpy based on 100% crystallized PLLA and stereocomplexed PLA (scPLA). Since the theoretical value is different for PLLA homocrystallites (α) (106 J∙g^−1^) and stereocomplexed crystallites (η) (142 J∙g^−1^) [44], it can be postulated that Δ*H*^0^_(h+sc)m_ value varies with relative amount of both α and η crystallite forms as follows:(3)$$\Delta H_{\left(h+sc\right)m}^{0}=\Delta H_{hm}^{0}\cdot X_{h}+\Delta H_{scm}^{0}\cdot X_{sc}\;\;\;\left[J\cdot g^{-1}\right],$$
where Δ*H*^0^_scm_ is melting enthalpy of 100% crystallized scPLA (142 J∙g^−1^). Values of *X*_h_ and *X*_sc_ are then relative amounts of α and η crystallites developed during non-isothermal crystallizations and can be calculated based on enthalpy values from DSC scans (see Table X) in the following manner.
(4)$$X_{h\;}=\frac{\Delta H_{hm}}{\Delta H_{hm}+\Delta H_{scm}},$$
(5)$$X_{sc\;}=\frac{\Delta H_{scm}}{\Delta H_{hm}+\Delta H_{scm}}$$

### 2.4. Mechanical Measurement

Tensile strength (*σ*_m_), elongation at break (*ε*_tb_), and Young’s modulus (*E*_t_) were measured by using a TIRA test 2300 (Tira, Schalkau, Germany) universal testing machine equipped with a load cell of 10 kN and extensometer MFX 500-B (Mess & Feinwerktechnik, GmbH, Velbert, Germany). Measurements have been performed according to the ISO 527 standard. Tensile strength and elongation at break measurements were performed at a crosshead speed of 5 mm∙min^−1^ and Young’s modulus at a crosshead speed of 1 mm∙min^−1^. All the samples were prior to testing conditioned in a KSP climatic chamber (Teseco, Kostelec nad Orlici, Czech Republic) according to ISO 291 at 23 °C and 50% relative humidity for 4 days. Each batch of 1B type dumbbell specimens was subjected to 10 repetitive tests under an ambient temperature of 23 °C.

### 2.5. Impact Resistance

Charpy impact strength (*a*_cU_) was measured using a Resil 5.5 (Ceast, Pianezza, Italy) testing machine according to ISO 179–1/1eU standard. A pendulum with the nominal energy of 5 J and 2.9 m∙s^−1^ striking velocity was used. Unnotched samples (80 × 10 × 4 mm^3^) were used, and each batch was subjected to 10 repetitive tests under an ambient temperature of 23 °C.

### 2.6. Thermo-Mechanical Analysis

Heat deflection temperature (HDT) measurements were conducted according to ISO 75–2 standard on HDT/Vicat 6–300 Allround (Zwick/Roell, Ulm, Germany). Method A using a flexural stress of 1.8 MPa was applied, and each batch of samples (80 × 10 × 4 mm^3^) was subjected to 5 repetitive tests at a heating rate of 120 °C∙h^−1^.

Vicat softening temperature (VST) measurements were conducted according to ISO 306 using the above mentioned equipment. Method B120 using a force of 50 N and a heating rate of 120 °C∙h^−1^ was applied on each batch of samples (4 mm thick grip section of dumbbell specimens). Samples were subjected to 5 repetitive tests. 

### 2.7. Morphological Characteristic

Morphology of fractured surfaces was examined by field emission scanning electron microscopy (FE-SEM) TESCAN MIRA 3 (Tescan, Brno, Czech Republic) with an accelerated voltage of 2 kV. Unnotched samples were frozen overnight (−50 °C) and then crushed using a Resil 5.5 (Ceast, Pianezza, Italy) impact testing machine. A pendulum with the nominal energy of 5 J and 2.9 m∙s^−1^ striking velocity was used. Fractured surfaces were coated with 2 nm of platinum/palladium (Pt/Pd) alloy (80/20) using a sputter coater LEICA EM ACE200 (Leica, Wetzlar, Germany).

## 3. Results and Discussion

### 3.1. Non-Isothermal Crystallization (DSC Measurements)

The prepared samples were studied by non-isothermal DSC analysis. During the first heating, the influence of the injection molding technological conditions and the promoting effect of nucleating agents (PDLA, HA) on the thermal properties and crystallization was investigated. The subtracted data obtained from the DSC measurements, including the calculated degree of crystallinity, are presented in Table 2 (SF) and Table 3 (TF). DSC thermograms of the samples injected into the cold (50 °C) and hot (100 °C) molds are shown in Figure 1 and Figure 2. Figure 1 shows the DSC curves of samples with PLLA and HA (1, 5, 10, and 15 wt.%) which were injected into cold (50 °C) and hot molds (100 °C). PLLA exhibits a characteristic glass transition temperature (*T*_g_), two exothermic peaks, which can be attributed to cold crystallization (*T*_cc_) and recrystallization before melting (*T*_rc_) and melting of homocrystalites (*T*_hm_). This observation indicates the capability of PLLA chains to undergo further rearranging and recrystallization before melting. Both of these resulted from a slow crystallization process as PLLA was unable to crystallize properly during cooling. Samples from cold and hot molds exhibit nearly the same trends, which signify that chosen conditions for neat PLLA from hot mold were insufficient for the significant increase in crystallinity degree (see Table 1 and Table 2). On the other hand, PLLA samples incorporated with HA showed enhancement of the crystallization rate at both injection molding conditions. Enhanced crystallization after the introduction of HA was also observed by Akindoyo et al. [36]. Nanocomposites from cold molds exhibited with increasing content of HA a shift of *T*_cc_ to lower temperatures as well as a decrease of Δ*H*_cc_. This clearly indicates the nucleation efficiency of HA. The degree of crystallinity of PLLA/15HA samples increased from 12.0 to 17.9% when compared to neat PLLA. Furthermore, in-mold annealing at 100 °C mold worked synergistically with increasing content of HA on the crystallization of PLLA. Hydrogen bonding interactions between PDLLA and HA has been reported by Zhou et al. [45]. Moreover, synergistic effects of chain dynamics and enantiomeric interaction on the crystallization in PDLA/PLLA mixtures has been observed by Lv et al. [46]. Therefore, the formation of multiple hydrogen bonding between the enantiomers of polylactide (CH_3_⋯O interactions) and their C=O group with the P–OH group of HA could be behind this phenomenon. However, further study to explore this phenomenon should be conducted. Samples containing 15 wt.% of HA resulted in crystallization of PLLA to the extent that no cold crystallization has been observed during heating. The degree of crystallinity of in-mold annealed PLLA/15HA samples increased from 16.2 to 44.4% when compared to neat PLLA. As shown in Figure 2, cold crystallization peaks of asymmetric PLLA/5PDLA blend and related nanocomposite samples from cold mold exhibited the same trend. When compared to neat PLLA, *T*_cc_ shifted from 100 °C to 89 °C and remained unchanged with increasing content of the HA. Asymmetric blend and PLLA/5PDLA/1HA nanocomposites showed approximately the same cold crystallization enthalpy. However, with a further increase of the HA concentration, the cold crystallization peak became slightly weaker. This observation indicates that the nucleation effect of HA was inhibited due to the formation of stereocomplex (SC) crystallites, which hindered the mobility of PLLA macromolecular chains. A decrease in macromolecular chain the mobility due to low PDLA concertation in PLLA was observed by Shi et al. [19]. Based on these results, Shi postulated that formed SC crystallites worked as the physical crosslinking sites where segments of a number of the core of PLLA chains are formed. Due to this, only a part of the PLLA chains participated in the SC crystallites, which resulted in limited mobility of those PLLA chains. Furthermore, Yamane et al. [17] observed that in the presence of SC crystallites can be melted PLLA polymer significantly reinforced, showing a strong strain hardening feature. This indicates that the reserved SC crystallites can also serve as a rheological modifier to improve the low melt strength of PLLA. Furthermore, samples from cold mold exhibited small exothermic recrystallization peaks due to previous cold crystallization. Besides, endothermic homocrystallite melting peaks (*T*_hm_) at around 175 °C were also observed, with small endothermic melting peaks of stereocomplexed crystallites (*T*_scm_) at around 220 °C. Enthalpy values (Δ*H*_scm_) of stereocomplexed crystallites varied only slightly and did not increase with a higher HA content. Therefore, the HA has no positive effect of promoting selective stereocomplexation. All of the in-mold annealed asymmetric PLLA/5PDLA blend and related nanocomposite samples fully crystallized during processing, and no cold crystallizations or recrystallizations has been observed. Enthalpy values of the in-mold annealed stereocomplexed crystallites remained unchanged with the incorporation of HA.

### 3.2. Mechanical Properties (Static Tensile and Charphy Impact Tests)

The graph in Figure 3 shows that the value of the tensile modulus (*E*_t_) increases with increasing hydroxyapatite content. For pure PLLA, the value of the tensile modulus after in-mold annealing is comparable to the samples originating from the cold mold. In-mold annealed samples based on PLLA matrix shows a slight increase in *E*_t_ when compared to samples from cold mold. Such an increase is related to a higher degree of crystallinity (see Table 2 and Table 3). Furthermore, Ko et al. [39] observed the same trend using unmodified nano HA (up to 15 wt.%) in PLA samples. In the case of asymmetric samples with 5 wt.% PDLA, the difference was more pronounced than for materials where the matrix contained only PLLA. The synergistic action of specific stereocomplex interactions and increased nucleation rate due to HA caused during the processing and development of the morphology an increase in the modulus of elasticity when compared to samples from cold mold. In-mold annealed PLLA/5PDLA/15HA samples show a ~28% increase of *E*_t_ when compared to neat PLLA. 

The resulting values of the tensile strength (*σ*_m_) are shown in Figure 4, which shows that in the case of samples originating from the cold mold, the value of the tensile strength hardly changes depending on the content of the HA. For the samples containing PLLA and 1% HA that were injected into the hot mold, it can be seen that there was a slight increase in the value of the tensile strength compared to pure PLLA. In the case of the PLLA/5PDLA/1HA samples from the hot mold, the difference in strength is 17% higher compared to PLLA/5PDLA. The strength values of the materials gradually decrease with contents from 5% HA onwards. This is most likely due to the higher HA content, which agglomerated into larger clusters (see SEM fracture analysis). These agglomerates acted as structural defects and caused the anticipated failure of the samples under tensile stress. Furthermore, Vadori et al. [47] have shown that increasing the mold temperature of PLA decreases the impact toughness and ductility of PLA.

Elongation at break (*ε*_tb_) plots (see Figure 5) showed the same trend; all in-mold annealed samples showed a sharp decrease in ductility. Furthermore, PLLA based samples from cold molds show an obvious decrease of *ε*_tb_ with increasing content of HA. Lower ductility of cold mold samples has also been observed after blending PLLA with 5% of PDLA. 

From the plots in Figure 6 it can be seen that the in-mold annealed specimens with PLLA matrix have higher impact toughness (*a*_cU_) compared to the specimens from the cold mold. The highest values were observed for in-mold annealed PLLA/1HA nanocomposites. The same trend has been observed by Kawamoto et al. [48] using different nucleation agents. High-temperature molding (110 °C) of PLLA and nucleating agents based on a mix of ethylenebis-(12-hydroxystearylamide) (EBHS)/talc (1 wt.% each) and octamethylenedicarboxylicdibenzoylhydrazide (OMBH) at 1 wt.% loading. Furthermore, the value of impact toughness decreased with an increasing percentage of HA in PLLA. A similar trend of the impact toughness was observed for the samples originating from the hot mould with a base matrix containing 5 wt.% PDLA. 

An exception is the trend of values for samples from the cold mould containing PDLA. The highest impact toughness value has been observed for the pure biopolymer without adding HA component and onward decreases with increasing content. Enhanced impact toughness of annealed PLA with 1 wt.% of ethylenebishydroxystearamide (EBH) has been observed by Tang et al. [49]. An increased number of spherulites with smaller size were believed to consume more energy and thus increase the impact strength of nucleated PLA samples.

### 3.3. Thermo-Mechanical Properties (HDT and VST Measurements)

The heat deflection temperature (HDT) and Vicat softening temperature (VST) of cold and hot molded samples are shown in Figure 7 and Figure 8, respectively. VST is the temperature at which a material loses its stability form, and HDT is the temperature at which the material loses its load-bearing capacity. Both HDT and VST results of cold mold samples proved that a degree of crystallinity below 20% (see Table 2 and Table 3) is insufficient to enhance thermal stability. The same results were reported by Ferri et al. [41]. Tang et al. [49] noticed enhanced HDT of PLA with 1% EBH molded at room temperature and then annealed for 1, 2, 4, 10, and 20 min at 105 °C. A threshold for crystallinity content was noticed when the crystallinity reached 25%. Our in-mold annealed samples show the same trend. However, this phenomenon is more obvious from VST results where an in-mold annealed PLLA/5HA sample at 25.4% crystallinity reached VST of 75.6 °C and a cold mold sample at 15.6% crystallinity reached only 61.4 °C. The highest HDT increase of in-mold annealed samples without PDLA was noticed by 13% for PLLA/15HA and by 24% for PLLA/5PDLA/15HA samples when compared to hot mold neat PLLA. 

The same trend was noticed for VST, where PLLA/15HA increased by 37% and PLLA/5PDLA/15HA by 44%. These results prove that the developed morphology during in-mold annealing at 100 °C for 2 min worked synergistically, and enhanced temperature resistance was reached due to specific stereocomplex interactions at a low loading of PDLA (5 wt.%) and relatively high HA loading (15 wt.%).

### 3.4. Fracture Surface Morphology (SEM Analysis)

SEM analysis was used to evaluate the morphology of individual fracture surfaces of PLLA, asymmetric PLLA/5PDLA blends and their nanocomposites with different HA content. In particular, the influence of the preparation technology on the dispersion and distribution of the nanofiller and the fracture mechanism of the prepared samples were investigated. The following images show the PLLA samples that were prepared by injection moulding in the cold (Figure 9a) and hot (Figure 9b) molds. It is well known that in many applications, the fracture properties of amorphous polymers render them unsatisfactory. These fracture properties are linked to the stress-induced growth and breakdown of crazes, which are planar, crack-like defects [50]. In the case of amorphous PLLA, the primary deformation mechanism propagation comes from multiple craze formations with low initiation energy. On the other hand, in crystalline PLLA, deformation under impact loading takes place through the deformation of the crystallites with the contribution of cavitation and fibrillation mechanisms [51]. Figure 9 shows that cold mold PLLA sample with 12% crystallinity exhibits smoother surface fracture when compared to in-molded annealed PLLA with higher roughness and 16.2% crystallinity. Both details of fractured surfaces show few fibrils. Since the higher impact toughness was observed for in-mold annealed PLLA (see Figure 6), an increase of spherulites with smaller size most likely consumed more energy during fracture. An increase in surface roughness was assigned by Park et al. [52] to the increase of crystallinity. The fracture surface morphology of asymmetric PLLA/5PDLA blends, which were prepared by injection moulding in cold and hot molds is shown in Figure 10a,b, respectively. Both cold and hot mold samples similarly show brittle fracture failure without any indication of plastic deformation. The coarser failure structure with deep concavities of the in-mold annealed samples reflects a higher degree of crystallinity (see Table 2 and Table 3). However, the impact toughness of these samples is much lower compared to the cold mold samples. Related to the spherulite structure of crystalline polymers, two possible crack paths should be considered: one is the inter-spherulitic crack growth, and the other the crack growth through spherulites. The polymer crystal structure related crack paths strongly depend on the formation process of the microstructure [53,54]. A more detailed study is required to clarify the effect of PDLA on homocrystallization and the formation of stereocomplexed structure, which resulted in subsequent failure and different crack propagation of cold and hot molded asymmetric PLLA/5PDLA blends.

The morphology of sample fractures with 1 wt.% HA (PLLA/1HA) is shown in Figure 11a (cold mold) and Figure 11b (hot mold). Similar to the previous samples, a coarser failure with deep concavities can be observed for in-mold annealed samples, reflecting a higher degree of crystallinity and torturous path progression of fracture fronts. Furthermore, both images show relatively good dispersion and distribution of HA (white dots) with just a few agglomerates in the PLLA matrix. In-mold annealed samples show a higher degree of cavitation in the nanofiller region and ductile fibrillation of PLLA. The existence of drawing fibrils and cavitation on the impact fracture surface confirms enhanced impact toughness (see Figure 6). An increase of the temperature in the crack-tip region above the glass transition temperature due to high strain-rate could be behind this kind of fibril formation [55]. Fine dispersion and distribution of HA resulted due to heterogeneous nucleation effect and in-mold annealing in the enhanced microstructure of PLLA/1HA nanocomposite. Furthermore, a possible explanation behind cavitation in HA region could be the hydrogen bonding, which was observed by Zhou et al. [45]. In their study, the IR and XPS analysis showed the formation of hydrogen bonding between the C=O group of PDLLA and the surface P–OH group of HA in the PDLLA/HA nanocomposites.

Asymmetric PLLA/5PDLA mixtures with 1 wt.% HA are shown in Figure 12a (cold mold) and Figure 12b (hot mold). The details of nanocomposite figures show relatively good dispersion of the nanofiller for both samples. The fracture surface of PLLA/5PDLA/1HA from cold mold has a similar fractured morphology to that of cold molded PLLA/5PDLA. In contrast, when comparing in-mold annealed PLLA/5PDLA/1HA and PLLA/5PDLA samples, the nanocomposite shows very different fracture morphology after the addition of 1 wt.% HA. Fracture curves of nanocomposite sample indicate a one-plane fracture, which is perpendicular to the applied load. Furthermore, PLLA/5PDLA/1HA has a less uneven progression of fracture fronts when compared to PLLA/5PDLA. In-mold annealed PLLA/5PDLA/1HA show an increase in impact resistance when compared to the samples without HA. On the other hand, the cold mold nanocomposite specimens show a decrease in impact toughness with the addition of 1 wt.% of HA, when compared to asymmetric blends. When comparing crystallinity degrees of nanocomposites (1 wt.% HA) and blends from both cold and hot molds they are nearly identical. Loose of tight entanglements of macromolecular chains at amorphous fractions due to stereocomplexation and homocrystallization could cause this phenomenon. 

The failure morphology of cold and hot molded samples with 15 wt.% HA (PLLA/15HA) is shown in Figure 13a,b, respectively. The fracture surface of PLLA/15HA from the cold mold shows typical brittle fracture failure of PLLA. The detail of the structure then shows the agglomeration of HA into higher structural units and a large number of cavities where agglomerates were tearing apart by the progression of fracture fronts. Agglomeration of unmodified HA within PLA matrix due to their higher surface energy of nanoparticles has been observed by Ko et al. [39]. In contrast, detail of in-mold annealed sample shows a more compact structure of the fractured surface. The different degrees of crystallinity (see Table 2 and Table 3) of the cold (~26%) and hot (~51%) molds do not reflect the previous trend of increasing impact toughness, which remains identical. Thus, the primary mechanism that led to the failure of the samples was the agglomeration of nanofiller into higher structural units that acted as local stress concentrators.

Fractured surfaces of cold (Figure 14a) and hot (Figure 14b) molded nanocomposites with 5 wt.% of PDLA, and 15 wt.% of HA showed similar failure mode as PLLA/15HA nanocomposites. The higher concentrations of nanoparticles of HA, while imparting good thermal resistance (see Figure 7 and Figure 8) to the biocomposite, induced lower mechanical properties (except tensile elastic modulus) at higher concentrations. 

## 4. Conclusions

In this work, the effect of in-mold annealing on the properties of nanocomposite systems based on asymmetric poly(l-lactide) (PLLA)/Poly(d-lactide) (PDLA) blends at 5 wt.% PDLA loading, which were incorporated with nano-hydroxyapatite (HA) at various concentrations (1, 5, 10, 15 wt.%) was presented and discussed. In-mold annealing at 100 °C mold for 2 min after injection molding was compared to relatively “cold” molded samples at a temperature of 50 °C. From the results, it was noticed that the tensile modulus, crystallinity, and thermal resistance were enhanced with increasing content of HA and blending with 5 wt.% of PDLA. In-mold annealing further enhanced the above-mentioned properties due to the synergistic action of specific stereocomplex interactions and increased nucleation rate due to the introduction of HA. In-mold annealed PLLA/5PDLA/15HA samples show an increase of the tensile modulus by ~28%; VST by ~44%; and crystallinity by ~59%, when compared to neat hot molded PLLA. On the other hand, the % elongation values at the break as well as the tensile strength of the PLLA and asymmetric nanocomposites were lowered with increasing HA content and in-mold annealing. In the case of asymmetric blends and related nanocomposites an embrittlement of material systems has been linked to in-mold annealing. Loose of tight entanglements of macromolecular chains at amorphous fractions due to stereocomplexation and homocrystallization could cause this phenomenon. On the other hand, impact toughness, when compared to neat cold-molded PLLA was improved by ~44% with in-mold annealing of PLLA/1HA. Furthermore, fracture morphology revealed fine dispersion and distribution of HA at 1 wt.% concentration. On the other hand, fractured surfaces with a high concentration of HA (15 wt.%) show agglomerates that worked as stress concentrators during impact loading. This is a clear sign that at higher concentrations of HA, prolonged melt-blending is insufficient to achieve good dispersion and distribution of HA within both PLLA and PLLA/5PDLA blends.

## Figures and Tables

**Figure 1 polymers-13-02835-f001:**
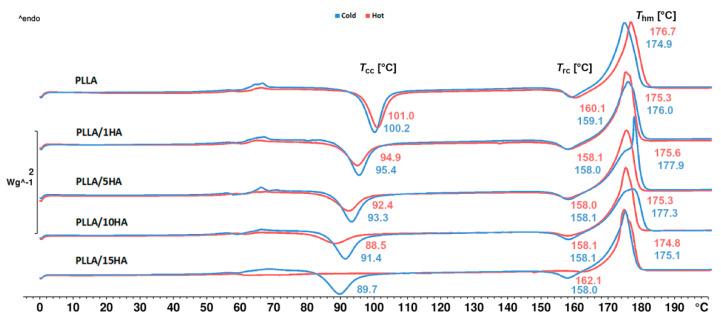
DSC curves of the PLLA a PLLA/HA nanocomposites obtained during the first heating.

**Figure 2 polymers-13-02835-f002:**
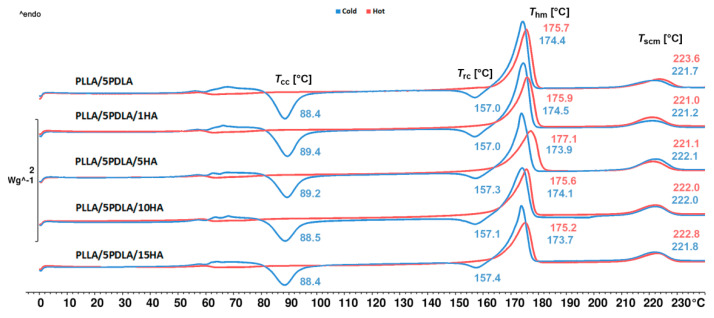
DSC curves of the asymmetric PLLA/5PDLA a PLLA/5PDLA/HA nanocomposites obtained during the first heating.

**Figure 3 polymers-13-02835-f003:**
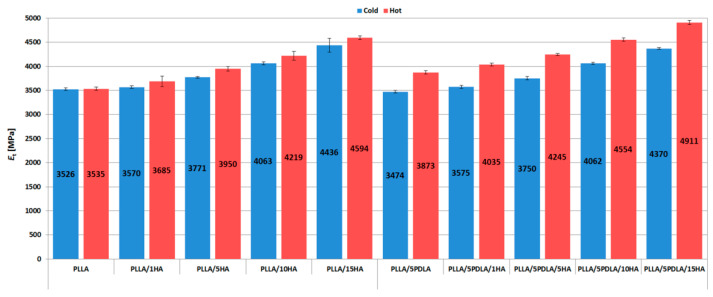
Tensile modulus plots of samples from cold and hot molds.

**Figure 4 polymers-13-02835-f004:**
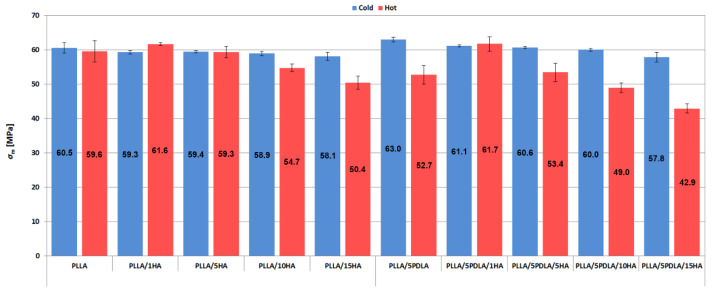
Tensile strength plots of samples from cold and hot molds.

**Figure 5 polymers-13-02835-f005:**
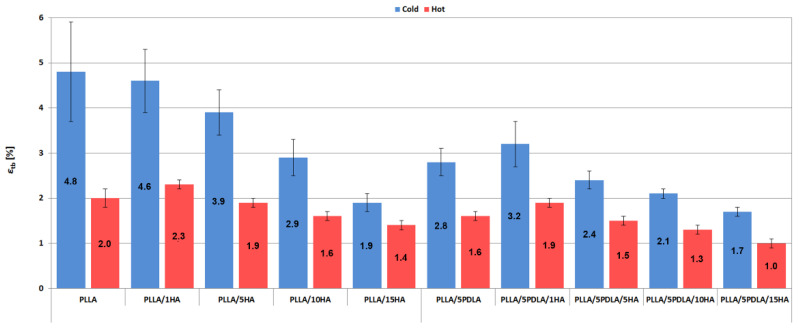
Elongation at break plots of samples from cold and hot molds.

**Figure 6 polymers-13-02835-f006:**
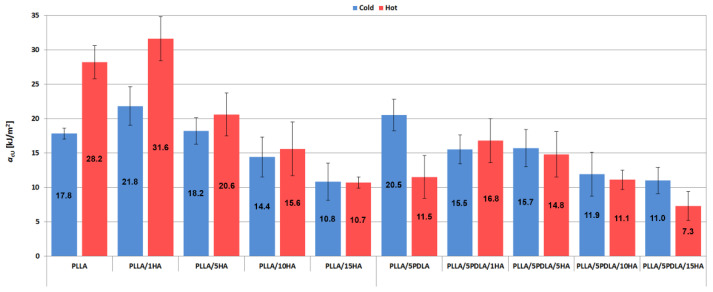
Impact toughness plots of samples from cold and hot molds.

**Figure 7 polymers-13-02835-f007:**
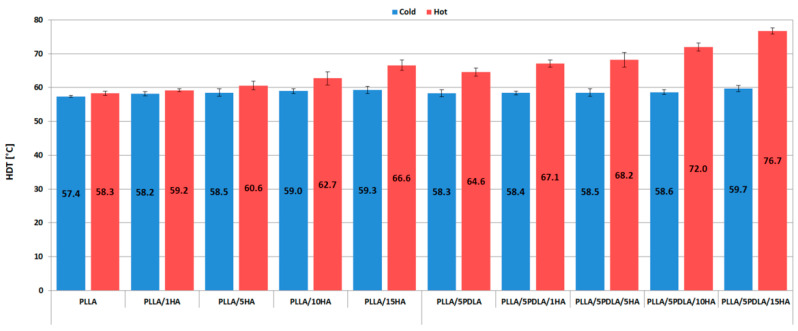
Heat deflection temperature plots of samples from cold and hot molds.

**Figure 8 polymers-13-02835-f008:**
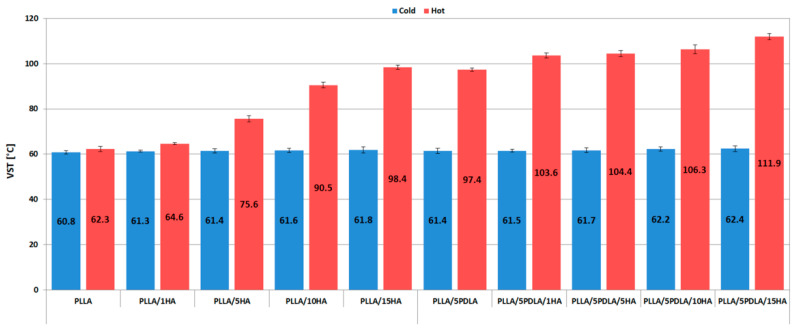
Vicat softening temperature plots of samples from cold and hot molds.

**Figure 9 polymers-13-02835-f009:**
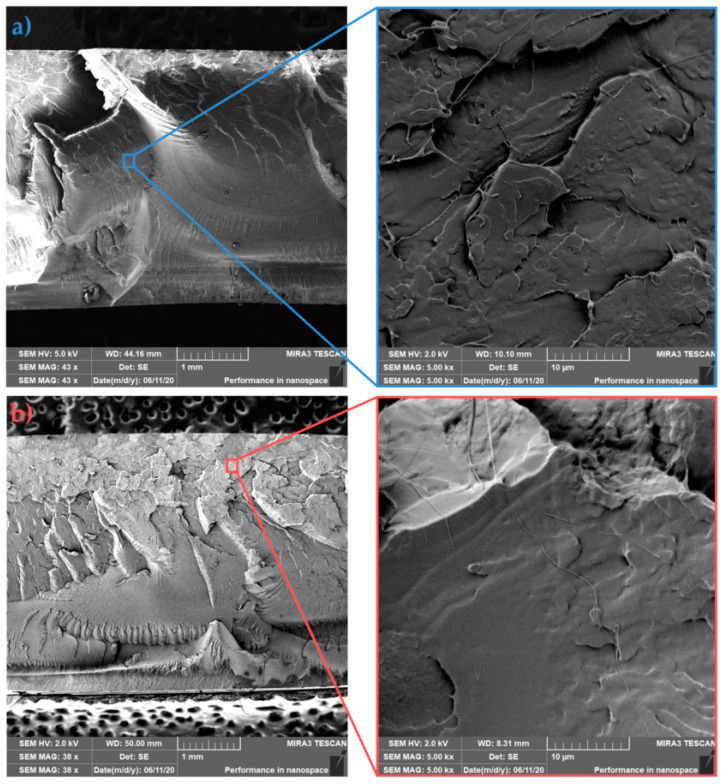
Fracture morphologies of (**a**) cold and (**b**) hot mold PLLA.

**Figure 10 polymers-13-02835-f010:**
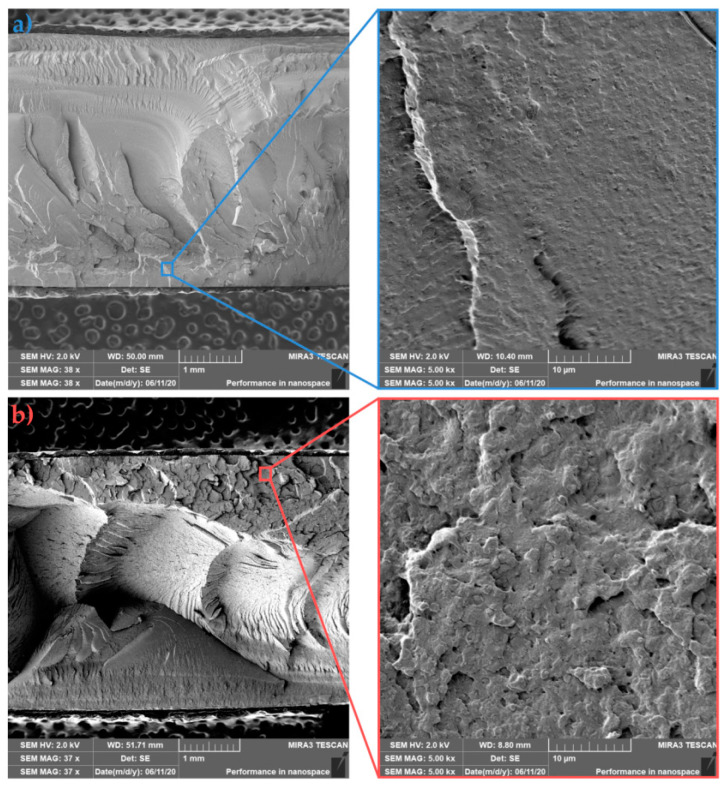
Fracture morphologies of (**a**) cold and (**b**) hot mold PLLA/5PDLA.

**Figure 11 polymers-13-02835-f011:**
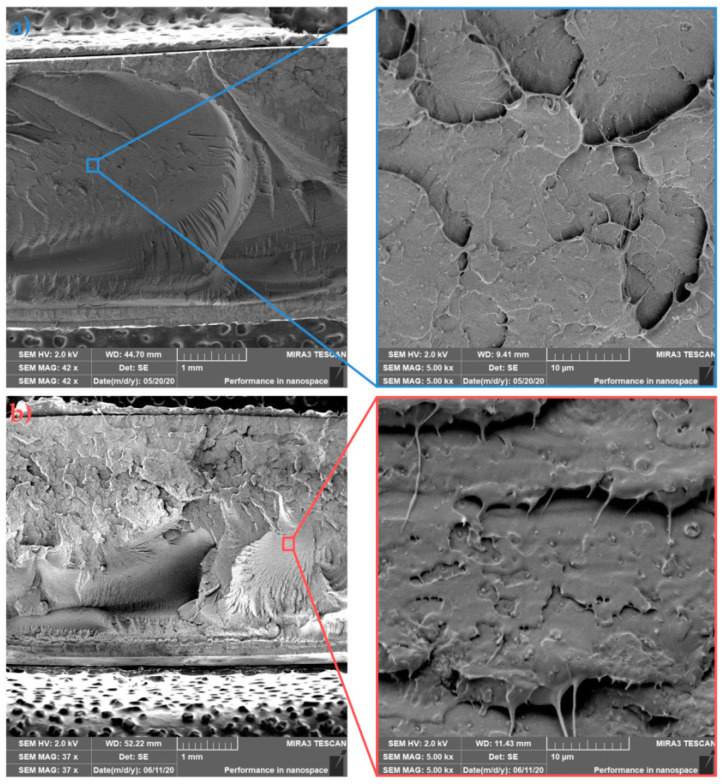
Fracture morphologies of (**a**) cold and (**b**) hot mold PLLA/1HA.

**Figure 12 polymers-13-02835-f012:**
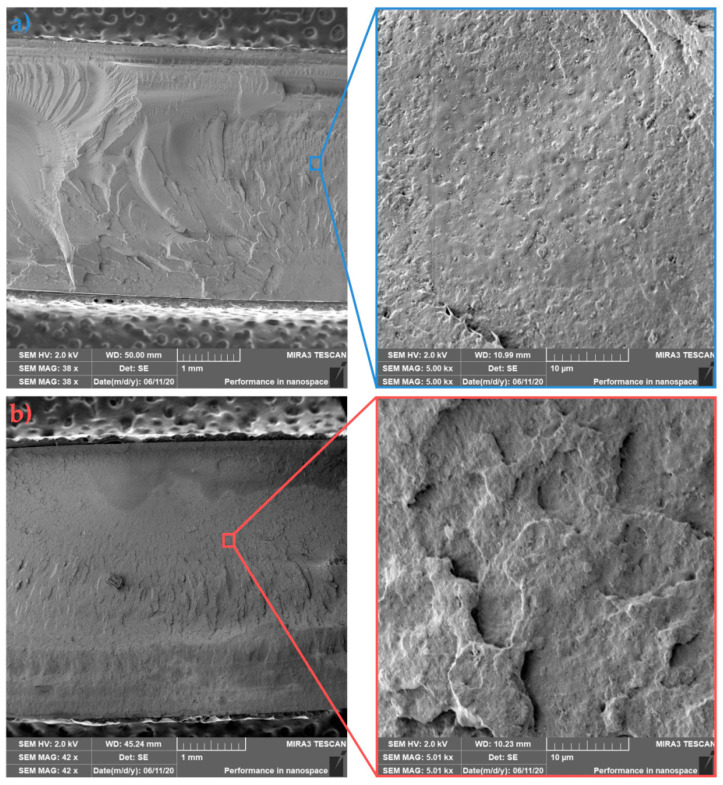
Fracture morphologies of (**a**) cold and (**b**) hot mold PLLA/5PDLA/1HA.

**Figure 13 polymers-13-02835-f013:**
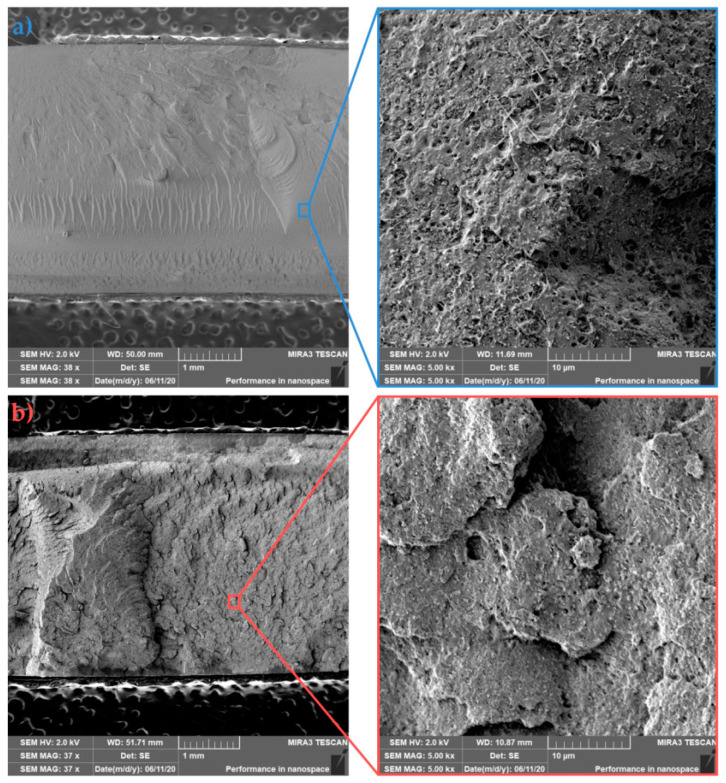
Fracture morphologies of (**a**) cold and (**b**) hot mold PLLA/15HA.

**Figure 14 polymers-13-02835-f014:**
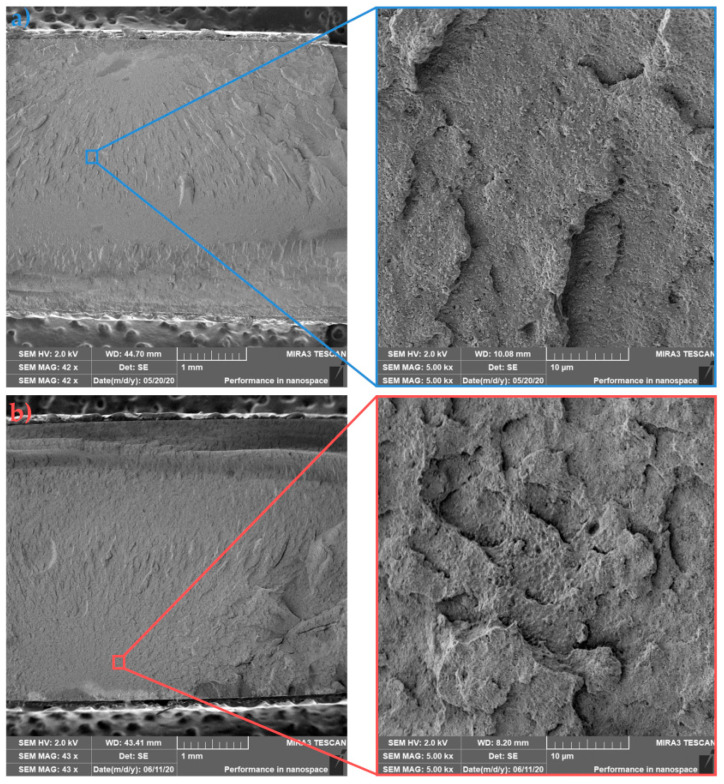
Fracture morphologies of (**a**) cold and (**b**) hot mold PLLA/5PDLA/15HA.

**Table 1 polymers-13-02835-t001:** Sample compositions.

Sample Code	PLLA (wt.%)	PDLA (wt.%)	HA (wt.%)
PLLA	100	-	-
PLLA/1HA	99	-	1
PLLA/5HA	95	-	5
PLLA/10HA	90	-	10
PLLA/15HA	85	-	15
PLLA/5PDLA	95	5	-
PLLA/5PDLA/1HA	94	5	1
PLLA/5PDLA/5HA	90	5	5
PLLA/5PDLA/10HA	85	5	10
PLLA/5PDLA/15HA	80	5	15

**Table 2 polymers-13-02835-t002:** Non-isothermal crystallization data of samples from cold mold (50 °C).

Samples	*T*_cc_ (°C)	Δ*H*_cc_ (J∙g^−1^)	*T*_rc_ (°C)	Δ*H*_rc_ (J∙g^−1^)	*T*_hm_ (°C)	Δ*H*_hm_ (J∙g^−1^)	*T*_scm_ (°C)	Δ*H*_scm_ (J∙g^−1^)	*χ_c_* (%)
PLLA	100.2	33.7	159.1	5.7	174.9	52.2	-	-	12.0
PLLA/1HA	95.4	29.5	158.0	7.0	176.0	49.3	-	-	13.7
PLLA/5HA	93.3	25.8	158.1	6.0	177.9	47.5	-	-	15.6
PLLA/10HA	91.4	24.9	158.1	5.6	177.3	46.1	-	-	16.4
PLLA/15HA	89.7	22.4	158.0	4.5	175.1	43.1	-	-	17.9
PLLA/5PDLA	88.4	23.1	157.0	4.4	174.4	44.5	221.7	8.6	16.1
PLLA/5PDLA/1HA	89.4	23.8	157.0	4.6	174.5	43.9	221.2	7.6	14.8
PLLA/5PDLA/5HA	89.2	20.1	157.3	3.4	173.9	38.0	222.1	11.5	14.5
PLLA/5PDLA/10HA	88.5	20.6	157.1	3.2	174.1	38.2	222.0	12.8	15.1
PLLA/5PDLA/15HA	88.4	19.1	157.4	2.5	173.7	35.9	221.8	9.5	15.8

**Table 3 polymers-13-02835-t003:** Non-isothermal crystallization data of samples from hot mold (100 °C).

Samples	*T*_cc_ (°C)	Δ*H*_cc_ (J∙g^−1^)	*T*_rc_ (°C)	Δ*H*_rc_ (J∙g^−1^)	*T*_hm_ (°C)	Δ*H*_hm_ (J∙g^−1^)	*T*_scm_ (°C)	Δ*H*_scm_ (J∙g^−1^)	*χ_c_* (%)
PLLA	101.0	26.5	160.1	6.0	176.7	49.7	-	-	16.2
PLLA/1HA	94.9	20.4	158.1	5.7	175.3	49.1	-	-	22.0
PLLA/5HA	92.4	17.1	158.0	4.7	175.6	47.4	-	-	25.4
PLLA/10HA	88.5	11.0	158.1	2.9	175.3	45.2	-	-	32.8
PLLA/15HA	-	-	162.1	0.5	174.8	40.5	-	-	44.4
PLLA/5PDLA	-	-	-	-	175.7	42.5	223.6	9.8	40.1
PLLA/5PDLA/1HA	-	-	-	-	175.9	39.9	221.0	9.6	38.1
PLLA/5PDLA/5HA	-	-	-	-	177.1	38.6	221.1	9.8	38.3
PLLA/5PDLA/10HA	-	-	-	-	175.6	37.1	222.0	9.6	38.9
PLLA/5PDLA/15HA	-	-	-	-	175.2	35.5	222.8	9.3	39.4

## Data Availability

Not applicable.

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
