# Peer review of "Effect of In-Mold Annealing on the Properties of Asymmetric Poly(l-lactide)/Poly(d-lactide) Blends Incorporated with Nanohydroxyapatite"

_polymers, 2021, doi:10.3390/polym13162835_

Round 1

Reviewer 1 Report

Reviewers' comments:

Manuscript Number: polymers-1349127

Full Title: Effect of In-mold Annealing on the Properties of Asymmetric Poly(L-lactide)/Poly(D-lactide) Blends Incorporated with Nanohydroxyapatite.

Comments: 

Overall, this paper meets the journal's level of publication. But there are some major issues that need to be addressed. I suggest that the paper be published after major revisions.

- In the Abstract – too long, the authors need to improve with more specific short results and conclusions.

- Add more suitable keywords.

- The introduction section should be improved; more related papers must be discussed and superiority, novelty, critical improvement in this study must be clarified.

- The experimental section should be detailed especially for the Mechanical Analysis and Impact resistance.

- In part FESEM: how the energy of the accelerator beam used?

- Please provides the references for all equations and formula.

- Figures 3-5, is not clear make clear.

- The conclusion part should rebuild to let it fluent.

- References: there are recent references in 2021 treating the same subject, you can use and make all references in same format for volume number, page numbers and journal name, because it is difficult to searching and reading.

- Language needs substantial improvement. Please consult a native English speaker or a language editing service.

Author Response

In the Abstract – too long, the authors need to improve with more specific short results and conclusions.

Abstract has been shortened and more specific results added.

Add more suitable keywords.

Keywords have been added and updated.

The introduction section should be improved; more related papers must be discussed and superiority, novelty, critical improvement in this study must be clarified.

The introduction section has been updated with more actual and recent papers. Furthermore, the novelty section has been extended to show more emphasis on the newly achieved results.

The experimental section should be detailed especially for the Mechanical Analysis and Impact resistance.

Specifications of mechanical analysis and impact resistance at the experimental section have been extended.

In part FESEM: how the energy of the accelerator beam used?

The same parameters as for impact resistance have been used: a pendulum with the nominal energy of 5 J and 2.9 m∙s−1 striking velocity. The text has been updated.

Please provide the references for all equations and formulas.

Updated.

Figures 3-5, is not clear make clear.

Higher-quality images have been uploaded.

The conclusion part should rebuild to let it fluent.

The conclusion has been rewritten.

References: there are recent references in 2021 treating the same subject, you can use and make all references in the same format for volume number, page numbers and journal name, because it is difficult to searching and reading.

More recent literature has been added and discussed.

The format of references should be according to Polymers template.  

Language needs substantial improvement. Please consult a native English speaker or a language editing service.

Checked by a native speaker.

Reviewer 2 Report

The present study reports on Effect of In-mold Annealing on the Properties of Asymmetric Poly(L-lactide)/Poly(D-lactide) Blends Incorporated with Nanohydroxyapatite, and a group of experimental results have been collected to support it. After carefully reading it, the following points are suggested to consider.

1. “2.4. Mechanical Analysis” is revised to “2.4. Mechanical Measurement”; “2.6. Morphological Analysis” is revised to “2.6. Morphological Characteristic”;

2. peak values in figures 1 and 2 are suggested to add to easily catch and compare.

3. the working principle of PDLA and HA has not clearly presented, how about the synergistic effect of PDLA and HA? There is no explanation for it, I suggest to consider to discuss on it, the following references may be useful to explain about the synergistic effect, i.e., (1) Erik H. Weber, Matthew L. Clingerman, Julia A. King. Thermally conductive nylon 6,6 and polycarbonate based resins. I. Synergistic effects of carbon fillers. Journal of Applied Polymer Science. 2003, 88, 1, 112-122. (2) Haibao Lu and Wei Min Huang. Synergistic effect of self-assembled carboxylic acid-functionalized carbon nanotubes and carbon fiber for improved electro-activated polymeric shape-memory nanocomposite. Applied Physics Letters. 2013, 102(23): 231910.

In all, it is an interesting and useful study, it is worthy to be recommended.

Author Response

1)“2.4. Mechanical Analysis” is revised to “2.4. Mechanical Measurement”; “2.6. Morphological Analysis” is revised to “2.6. Morphological Characteristic”.

Topics have been rewritten according to suggestions.

2) Peak values in figures 1 and 2 are suggested to add to easily catch and compare.

Figures 1 and 2 have been updated with peak values.

3)The working principle of PDLA and HA has not clearly presented, how about the synergistic effect of PDLA and HA? There is no explanation for it...

The discussion of the synergistic effect of PLLA/PDLA and HA on the crystallization has been enhanced using the following literature:

(45)Zhou, Shaobing, et al. "Hydrogen bonding interaction of poly (D, L-lactide)/hydroxyapatite nanocomposites." Chemistry of materials 19.2 (2007): 247-253.

(46)Lv, Tongxin, et al. "Synergistic effects of chain dynamics and enantiomeric interaction on the crystallization in PDLA/PLLA mixtures." Polymer 222 (2021): 123648.

Round 2

Reviewer 1 Report

The authors have improved the revised manuscript significantly, I recommend acceptance.